# A Case Series on Pain Accompanying Photoimmunotherapy for Head and Neck Cancer

**DOI:** 10.3390/healthcare11060924

**Published:** 2023-03-22

**Authors:** Yuma Shibutani, Haruna Sato, Shinya Suzuki, Takeshi Shinozaki, Hayato Kamata, Kazuki Sugisaki, Atushi Kawanobe, Shinya Uozumi, Toshikatsu Kawasaki, Ryuichi Hayashi

**Affiliations:** 1Department of Pharmacy, National Cancer Center Hospital East, Kashiwa 277-8577, Japan; 2Department of Head and Neck Surgery, National Cancer Center Hospital East, Kashiwa 277-8577, Japan

**Keywords:** photoimmunotherapy, cetuximab sarotalocan sodium, transnasal endoscopy, locoregional recurrence of head and neck carcinoma

## Abstract

One of the most severe side effects of photoimmunotherapy (PIT) for head and neck cancer is pain. As there are presently no detailed reports on pain and pain management in PIT, we conducted a retrospective case series study. We conducted a retrospective study of five patients who had received PIT at the National Cancer Center Hospital East between January 2021 and June 2022 using medical chart data. All patients experienced pain, evidenced by an increased numerical rating scale (NRS) after PIT, regardless of the illumination method. The daily change in mean NRS rating shows that the pain was highest on the day of PIT, with ratings of 6.8 and 7.8 for the frontal and cylindrical diffuser methods, respectively; it dropped the following day quickly. Four of the five patients received fentanyl injections for postoperative pain management beginning on postoperative day (POD) 0. All patients who underwent therapy using a cylindrical diffuser required postoperative pain management with opioid drugs. Pain after PIT tended to be most intense immediately after or one hour after illumination and declined the following day, suggesting the need to have a pain relief plan in place in advance.

## 1. Introduction

Over 90% of head and neck cancers are squamous cell carcinomas (HNSCCs), and locally advanced HNSCC has a poor prognosis with a reported five-year survival rate of <50% [1,2]. The radical treatment option for unresectable, locally advanced HNSCC is chemoradiotherapy (CRT). Owing to the difficulties associated with re-irradiation, the preferred treatment for subsequent recurrence is systemic chemotherapy; however, this method has a low objective response rate [3,4,5,6].

Photoimmunotherapy (PIT) is a treatment in which cancer cells are selectively destroyed by injection of cetuximab sarotalocan sodium (a conjugate of the epidermal growth factor receptor (EGFR) antibody cetuximab, and a light-activated dye (IRDye700DX: IR700)), followed by non-thermal red light (690 nm) illumination of the tumor site using a frontal or cylindrical diffuser 20–28 h later. It is expected that irradiation with 690 nm laser light will activate the dye and induce a rapid cytotoxic reaction only on cells to which the complex has bound. Since light-induced activation and antigen-antibody binding are required for induction of the apoptotic effect, it is expected to selectively destroy only cancer cells while minimizing damage to the normal tissue surrounding the tumor cells. The mechanism of action of this therapy is that the activation of antibody complexes causes physical stress on the cell membrane, resulting in an increase in transmembrane water, which ultimately leads to cell rupture and necrosis. This reaction is considered to occur in a short period of time, as short as one minute after laser irradiation. It has also been suggested that the rapid release of immunogenic signals from cancer cells induces immature dendritic cells to mature and trigger a host immune response against the tumor, but this has not been clinically confirmed at this time.

As tumor illumination methods, cylindrical diffusers placed in needle catheters are used to treat interstitial tumors, while frontal diffusers are used to treat superficial tumors. (Figure 1) The non-thermal red light is applied to the tumor using a frontal diffuser for superficial light illumination for tumors ≤1 cm from the skin or mucosal surface or a cylindrical diffuser for interstitial illumination for tumors > 1 cm from the skin or mucosal surface. The illumination time for frontal and cylindrical diffusers is 5 min for each treated area. For interstitial illumination, cylindrical diffusers are placed uniformly into the tumor 1.8 ± 0.2 cm apart using 17-gauge closed-tipped needle catheters under radiographic or ultrasound imaging.

Cetuximab was selected for this treatment because 80–90% of HNSCC patients express EGFR [7,8,9,10]. In light of the results of phase I trials in Japan and phase I/II trials in the United States of America, PIT was conditionally approved in Japan in January 2021 for the treatment of unresectable locally advanced or locally recurrent head and neck cancer that cannot be treated with CRT or other standard therapies [7,11]. PIT is contraindicated in cases of tumor invasion into the carotid artery. In addition, patients with tumor invasion into the jugular vein or other organs may experience hemorrhage or tumor hemorrhage associated with tumor shrinkage or necrosis, so the decision to treat must be made carefully after careful consideration of the efficacy and risks of the therapy. The use of PIT as adjuvant therapy to other therapies is not indicated because its efficacy and safety have not been established at present.

One of the most severe side effects of PIT is pain. In the Japanese phase I trial, 100% of the patients reported pain at the application site, and pain of Grade 3 or above was common, being observed in 33.3% of patients. Therefore, pain management must be provided when PIT is used in medical practice [7]. Unfortunately, the number of patients who receive PIT is low because of PIT’s highly specific indication, namely, unresectable, locally advanced or recurrent head and neck cancer that was previously treated with CRT. As there are presently no detailed reports on pain and pain management in PIT, we conducted a retrospective case series study on these aspects.

## 2. Materials and Methods

### 2.1. Participants

We conducted a retrospective study of HNSCC patients who had received PIT at the National Cancer Center Hospital East between January 2021 and June 2022 using medical chart data. In this period, a total of six patients received PIT in the National Cancer Center Hospital East. However, one patient was excluded from the study because she was advanced in age and cognitively impaired, which made it difficult to assess pain correctly, and nurses could not evaluate her pain correctly, either. The five patients consisted of two men and three women. All patients had an Eastern Cooperative Oncology Group Performance Status (ECOG-PS) of <1 and were able to communicate their pain status articulately and without any issues. The median age was 60 years. Two patients had buccal mucosa cancer, two had oropharyngeal cancer, and one had nasopharyngeal cancer. Additionally, three of the five eligible patients received PIT more than once. Of these patients, one patient received a first PIT at another hospital, and one patient received the first PIT as part of a Phase I clinical trial, so the first data for these two patients were excluded from the analysis. Hence, we evaluated a total of nine PIT sessions, which received PIT at the National Cancer Hospital East as daily clinical practice, and the number of times PIT was performed in each of the five cases: once, twice, and three times in two, two, and one patient, respectively. Regarding the illumination technique, two patients received treatment via frontal diffuser only, one via cylindrical diffuser only, and the remaining two via both techniques (Table 1).

### 2.2. Analgesic Management and Pain Assessment

Postoperative pain management may be difficult for a surgeon alone, and consultation with a palliative care specialist is recommended if postoperative pain management is inadequate or pain symptoms are severe [12]. Additionally, the use of morphine and other opioids is often necessary for the postoperative setting [13]. Therefore, in this study, pain management was conducted on a case-by-case basis as part of general care by the attending physician in consultation with a supportive care team led by a palliative care doctor; no specific protocol or regimen was followed. Pain assessment was performed using the numerical rating scale (NRS) [14], a method routinely used in clinical practice, and the study was conducted using NRS data recorded in medical charts. 

This case series study was based on a daily clinical practice in which a nurse periodically asked the patient to rate their pain after receiving PIT, using the NRS, which comprises ten pain levels, with the 10th level denoting maximum pain; this information was recorded in the patient’s medical record. Pain after PIT was frequently assessed in all patients and recorded in the electrical medical records because it was recognized as an important symptom incurred by PIT.

### 2.3. Statistical Analysis

Descriptive statistics were used to analyze the means, medians, and frequencies. All calculations were performed using Microsoft Excel 2021 (Microsoft Corporation, Redmond, WA, USA).

### 2.4. Ethical Considerations

This case report was approved according to its compliance with the Ethical Guidelines for Life Sciences and Medical Research Involving Human Subjects and was subjected to the ethical review procedures of the National Cancer Center. Compliance with the relevant guidelines was also ensured while performing research involving the participants during the original studies. Ethical approval for this study was obtained from the National Cancer Center Institutional Review Board (Research Project No. 2021-262). The Institutional Review Board asserted that informed consent to the study was not required due to its retrospective chart review design.

## 3. Results

### 3.1. Pain after PIT

All patients experienced pain evidenced by increased NRS after PIT regardless of the illumination method. Patients 1, 3, and 4 reached the maximum NRS rating of 10 one hour after illumination. In Patient 4, for whom a frontal diffuser was used, NRS was mild (2–3) six hours after illumination, whereas in Patients 1 and 3, for whom a cylindrical diffuser was used, the pain persisted with NRS ratings of 6–9 six hours after illumination. 

The daily change in mean NRS rating shows that pain was highest on the day of PIT, with ratings of 6.8 and 7.8 for the frontal and cylindrical diffuser methods, respectively; it dropped the following day quickly. The NRS decreased slightly until the postoperative day (POD) 3. However, patients for whom the cylindrical diffuser method was used subsequently showed an increase in NRS at POD 4. This increase in NRS is not significantly different from the NRS of POD 1-3 and is similar to that before hospitalization (Figure 2) (Table 2).

### 3.2. Opioid Therapy for Pain after PIT

Four of the five patients received fentanyl injections for postoperative pain management beginning on POD 0. All patients who underwent therapy using a cylindrical diffuser required postoperative pain management with opioid drugs, but for one of the two patients who underwent therapy using a frontal diffuser, opioid pain management was deemed unnecessary and was thus not performed. 

In Patients 1 and 3, for whom a cylindrical diffuser was used, the maximum opioid dose required (morphine equivalent dose 94–330 mg/day) and the mean NRS (4–7.5) tended to be higher than in patients for whom a frontal diffuser was used. In both cases, the required opioid dose dropped from POD 2 onward and reached a dose comparable to the dose prior to treatment on POD 4 (Figure 3) (Table 3).

### 3.3. Fluctuations in Laboratory Test Results after PIT

No increase in body temperature or C-reactive protein (CRP) was observed in the frontal diffuser patients after PIT, but a slight increase in body temperature was noted in the cylindrical diffuser patients on POD 0 and POD 1. On POD 4, body temperature had returned to baseline, but CRP elevation was observed (Table 4).

## 4. Discussion

To our knowledge, this is the first case series study on pain and pain management after PIT. All patients experienced pain after PIT, which is consistent with the results of a phase I clinical trial in Japan [7]. Even though this study is a case series, the data obtained clearly described characteristics of pain caused by PIT in daily clinical practice. Concerning fluctuations in NRS, pain after PIT tended to be most intense immediately after or one hour after illumination and declined the following day, suggesting the need to have a pain relief plan in place in advance. A possible cause of the early onset of pain after PIT is the rapid destruction of IR700-bound tumor cells immediately after irradiation, resulting in irreversible morphological changes [15]. Therefore, preemptive analgesic planning is necessary before initiating PIT. 

NRS tended to be lower upon treatment with a frontal diffuser compared to that with a cylindrical diffuser. In general surgery, the pain has been reported to be mild in less invasive cases [16]. Cylindrical diffusers are more invasive than frontal diffusers because they can be used only with a needle catheter. Therefore, the use of frontal diffusers in this study may have resulted in lower NRS after PIT compared to cylindrical diffusers. With both illumination methods, opioid use resulted in a drop in NRS on POD 1. 

The required opioid dose had returned to baseline by POD 4, demonstrating that postoperative pain is a manageable symptom. NRS decreased slightly to POD 3 for both frontal and cylindrical diffusers.

However, the NRS increased again at POD 4 in the cylindrical diffuser. This increase in the NRS for POD 4 was similar to the baseline and may have been a patient-reported bias, as opioid use was also decreasing. Additionally, while NRS tended to be lower for treatment with a frontal diffuser than with a cylindrical diffuser, opioid pain management was still necessary, suggesting that opioid pain management is essential for PIT. However, it is important to monitor the pain status and management continuously, as there have been reports of pain continuing for more than 4 weeks after administering PIT [17]. From the phase I trial results, we understood that pain is a critical adverse event of PIT. Therefore, we asked the palliative care team, which consisted of palliative care doctors, nurses, pharmacists, and other medical staff, to manage the pain, and thus, we successfully controlled the adverse events in daily practice.

PIT has been reported to induce necrosis of the tumor site shortly after the illumination [15,16]. In the present study, no residual CRP was observed postoperatively in cases where a frontal diffuser was used. In contrast, residual CRP was observed after treatment in cases where a cylindrical diffuser was used. Early necrosis of the tumor site as a result of surgical invasion may be the cause of postoperative CRP elevation in these cases. In our study, postoperative infection was ruled out due to the absence of pyrexia and white blood cell elevation, suggesting that inflammatory symptoms from the illumination may have contributed to residual pain. Accordingly, it is necessary to continue to monitor pain status even after postoperative pain has improved. 

This study has two limitations. First, the pain symptoms and management could not be clarified due to the small sample size. Second, as the study is retrospective, the missing data values or reproducibility of the NRS could not be compensated for during the analysis. Since data were collected from medical records, missing data could not be evaluated. In addition, because the NRS assesses only the intensity of pain, it was difficult to identify the nature of the pain. Furthermore, the NRS is based on the patient’s subjective assessment, making it impossible to achieve reproducibility. Nevertheless, the NRS is commonly used in clinical practice to assess pain intensity, and in this study, pain management could be assessed by analyzing the variability of the NRS after PIT. More data needs to be collected to clarify the results of this study in more detail, and prospective data collection should be considered in the future.

## 5. Conclusions

In conclusion, although the pain accompanying PIT varies according to the method used, it develops early and can be successfully controlled using opioid analgesics with quick results. Thus, the importance of preemptively establishing a pain management plan was demonstrated. Going forward, it is necessary to further verify fluctuations in pain status and accompanying analgesic treatments through prospective clinical trials in order to establish postoperative pain management protocols for PIT.

## Figures and Tables

**Figure 1 healthcare-11-00924-f001:**
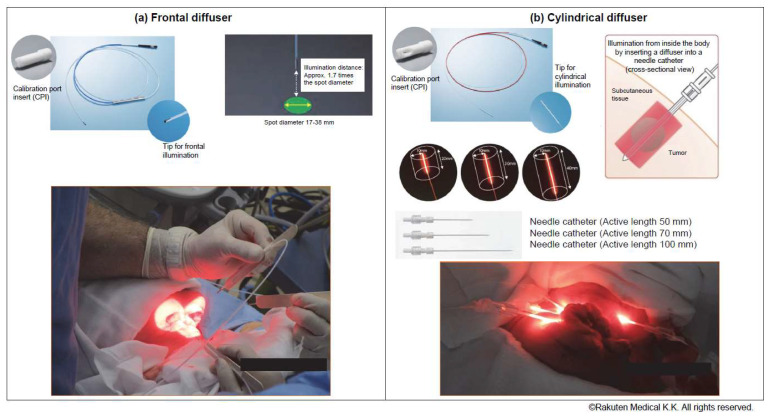
Illumination methods: A (**a**) frontal or (**b**) cylindrical diffuser. Cylindrical diffusers placed in needle catheters are used to treat subcutaneous or large tumors, whereas frontal diffusers are used to treat superficial tumors.

**Figure 2 healthcare-11-00924-f002:**
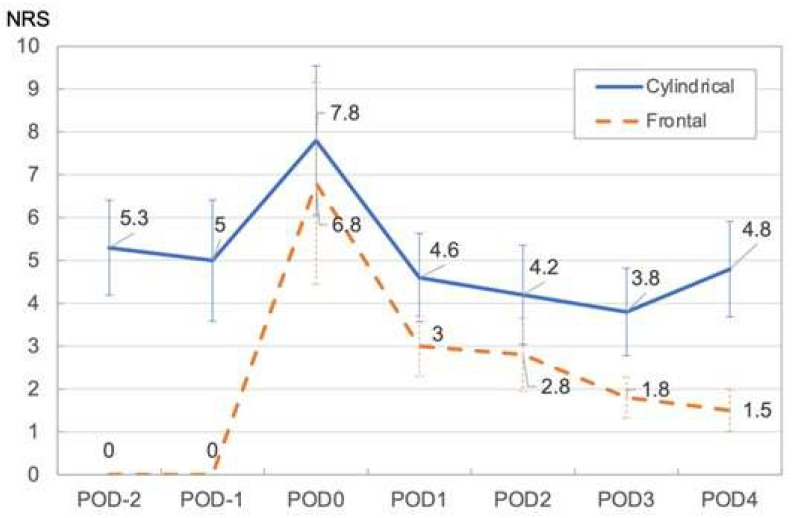
Mean NRS in Cylindrical and Frontal illumination method. Abbreviations: NRS: numerical rating scale, PIT: photoimmunotherapy, POD: postoperative day (days since treatment).

**Figure 3 healthcare-11-00924-f003:**
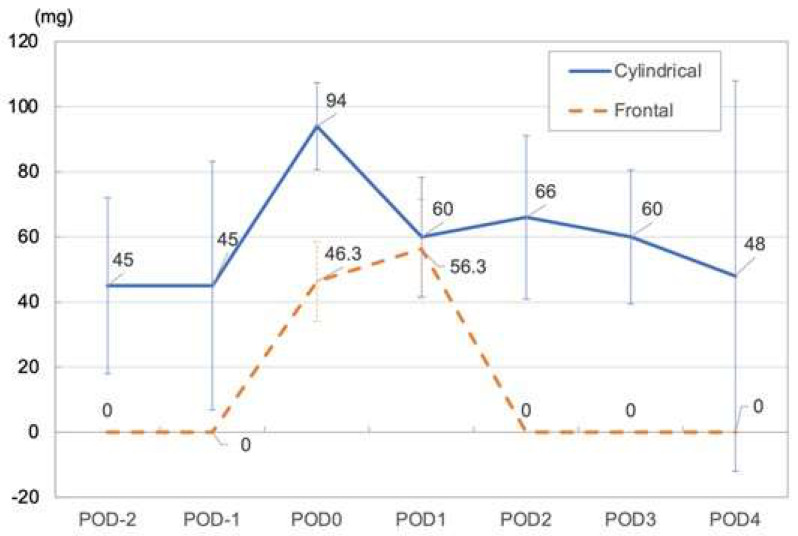
Mean opioid dose (Median morphine equivalent dose) in cylindrical and frontal illumination methods.

**Table 1 healthcare-11-00924-t001:** Patient characteristics.

		N = 5	%
Sex	Male	2	40
	Female	3	60
ECOG PS	0	4	80
	1	1	20
Age (years)	Median (range)	60.5 (51–74)	
Tumor	Buccal mucosa cancer	2	40
	Oropharyngeal cancer	2	40
	Nasopharyngeal cancer	1	20
Number of PIT sessions	1	2	40
	2	2	40
	3	1	20
Treatment method	Frontal + cylindrical	2	40
	Cylindrical	1	20
	Frontal	2	40

Sex, ECOG PS, age, and tumor are characteristics of the five patients, while the number of PIT sessions and treatment method are aggregated data obtained over nine sessions. Abbreviations: ECOG PS: Eastern Cooperative Oncology Group Performance Status, PIT: photoimmunotherapy.

**Table 2 healthcare-11-00924-t002:** NRS changes immediately after PIT.

Case	Session	Sex	Tumor Site	Method		NRS				
				Frontal	Cylindrical	Pre-Treatment	Immediately after Treatment	One Hour Post-Treatment	Three Hours Post-Treatment	Six Hours Post-Treatment
1	1	Male	Left buccal mucosa	◯	◯	3	10	NE	8	8
	2					NE	10	9	8	9
2		Female	Right buccal mucosa	◯	◯	1	0	0	0	1
3	1	Male	Oropharynx		◯	4	NE	NE	8	6
	2					7	NE	10	10	6
4	1	Female	Oropharynx	◯		NE	7	5	NE	3
	2					0	NE	10	4	NE
	3					NE	10	8	6	2
5		Female	Nasopharynx	◯		NE	0	NE	NE	NE

Abbreviations: NE: not evaluated, NRS: numerical rating scale, PIT: photoimmunotherapy. ◯: Illumination methods used for PIT.

**Table 3 healthcare-11-00924-t003:** Maximum opioid dose and NRS and mean NRS changes.

Case	Session	Sex	Tumor Site	Method		Maximum Opioid Dose (mg) *	Maximum NRS	Mean NRS **
				Via Frontal Diffuser	Via Cylindrical Diffuser			
1	1	Male	Buccal mucosa	◯	◯	94	10	4.0
	2					119	10	4.2
2		Female	Buccal mucosa	◯	◯	60	3	2.1
3	1	Male	Oropharynx		◯	330	8	6.0
	2					126	10	7.5
4	1	Female	Oropharynx	◯		62.5	7	2.8
	2					65	10	3.4
	3					52.5	10	3.2
5		Female	Nasopharynx	◯		0	3	1.0

* Median morphine equivalent dose. Abbreviations: NRS: numerical rating scale, PIT: photoimmunotherapy, POD: postoperative day (days since treatment)**.** Morphine equivalent dose, ** Mean daily maximum NRS during hospitalization. Abbreviations: NRS: numerical rating scale, PIT: photoimmunotherapy. ◯: Illumination methods used for PIT.

**Table 4 healthcare-11-00924-t004:** Clinical laboratory test results during PIT.

Method	Test Result Median	POD-2	POD-1	POD0	POD1	POD2	POD3	POD4
Via frontal diffuser	Body temperature (°C)	36.6	36.7	36.6	36.6	36.4	36.4	36.5
	WBC (10^2^/μL)	49	NE		110	NE		
	CRP (mg/dL)	0.03			0.08			
Via cylindrical diffuser	Body temperature	36.7	36.8	37.1	37.1	36.9	36.7	36.6
	WBC (10^2^/μL)	42						58
	CRP (mg/dL)	0.59						4.42

Abbreviations: CRP: C-reactive protein, NRS: numerical rating scale, PIT: photoimmunotherapy, POD: post-operative day, WBC: white blood cell.

## Data Availability

Not applicable.

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
