# Peer review of "A Case Series on Pain Accompanying Photoimmunotherapy for Head and Neck Cancer"

_healthcare, 2023, doi:10.3390/healthcare11060924_

Round 1

Reviewer 1 Report

Review Report

·     In this case series, the authors aimed to assess pain and pain management following photoimmunotherapy (PIT) for head and neck cancer.

·     Based on a retrospective analysis of 5 patients who had received PIT, the authors concluded that pain after PIT tended to be most intense immediately after or one hour after illumination and declined the following day, suggesting the need to have pain relief.

·     The paper is interesting, well-structured, and correctly organized. The authors have clearly worked hard to detail their study, but I have some comments:

POINTS OF STRENGTH

1.     Interesting topic.

2.     The results are ok.

POINTS OF WEAKNESS

1.     Retrospective analysis.

2.     Small sample size.

SPECIFIC COMMENTS

1.     Ethical statement is required in the main text.

2.     Participants: We conducted a retrospective study of five of the six HNSCC……….. One patient was excluded from the study……………. these two participants were therefore excluded from the initial data analysis………What was the actual number of patients and the number of excluded patients?

Author Response

Comment 1:

Ethical statement is required in the main text.

[Response]

Thank you for pointing this out. The following text has been added following sentences.

“2.4. Ethical Considerations

This case report was approved according to its compliance with the Ethical Guidelines for Life Sciences and Medical Research Involving Human Subjects and was subjected to the ethical review procedures of the National Cancer Center. Compliance with the relevant guidelines was also ensured while performing research involving the participants during the original studies Ethical approval for this study was obtained from the National Cancer Center Institutional Review Board (Research Project No. 2021-262). The Institutional Review Board asserted that informed consent to the study was not required due its retrospective chart review design.” (Page: 4, line: 126-134)

Comment 2: 

Participants: We conducted a retrospective study of five of the six HNSCC……….. One patient was excluded from the study……………. these two participants were therefore excluded from the initial data analysis………What was the actual number of patients and the number of excluded patients?

[Response]

Thank you for pointing this out. We added more detailed explanation about the excluded patient and PIT sessions as follows:

“In this period, total six patients received PIT in the National Cancer Center Hospital East, however, one patient was excluded from the study because she was advanced in age and cognitively impaired, which made it difficult to assess pain correctly and nurses could not evaluated her pain correctly, either.” (Page: 3, line: 84-87)

“Of these patients, one patient received a first PIT at another hospital, and one patient received the first PIT as part of the Phase I clinical trial, so the first data for these two patients were excluded from the analysis. Hence, we evaluated total of nine PIT sessions, which received PIT at the National Cancer Hospital East as daily clinical practice, and the number of times PIT was performed in each of the five cases: once, twice, and three times in two, two, and one patients, respectively.” (Page: 3, line: 92-100)

For more details please see the revised version manuscript.

Reviewer 2 Report

Reviewer comments for healthcare-2255210

The authors Shibutani, et.al. has attempted to study the pain, the most severe side effects of photoimmunotherapy (PIT) for head and neck cancer. As there are presently no detailed reports on pain and pain management in PIT, they conducted a retrospective case series study. They conducted a retrospective study of five patients who had received PIT at the National Cancer Center Hospital East between January 2021 and June 2022 using medical chart data. All patients experienced pain evidenced by increased the numerical rating scale (NRS) after PIT regardless of the illumination method. The daily change in mean NRS rating shows that pain was highest on the day of PIT, with ratings of 6.8 and 7.8 for the frontal and cylindrical diffuser methods, respectively; it dropped quickly the following day. Four of the five patients received fentanyl injections for postoperative pain management beginning on postoperative day (POD) 0. All patients who underwent therapy using a cylindrical diffuser required postoperative pain management with opioid drugs. Pain after PIT tended to be most intense immediately after or one hour after illumination and declined the following day, suggesting the need to have a pain relief plan in place in advance.

Major Comments:

1: For a case study it is ok to report a case.

2: They need to cite more literature, if Pain management was not done before.

3: This study is about only 5 cases. No clear statistical analysis has not been done to show effect is significant and real. I think more number of patients study is needed to report this case study.

Author Response

Comment 1:

For a case study it is ok to report a case.

[Response]

Thank you for your advice. Since each case in this case report has a different tumor site and different surgical technique, the data was summarized to determine at what time of the year more pain was experienced and how much opioids were used in terms of morphine equivalents.

Comment 2:

They need to cite more literature, if Pain management was not done before.

[Response]

Thank you for your advice. The following text has been added.

“Postoperative pain management may be difficult for a surgeon alone, and consultation with a palliative care specialist is recommended if postoperative pain management is inadequate or pain symptoms are severe [12]. Also, the use of morphine and other opioids is often necessary in the postoperative setting [13]. Therefore, in this study, the pain management was conducted on a case-by-case basis as part of general care by the attending physician in consultation with a supportive care team led by a palliative care doctor; no specific protocol or regimen was followed.” (Page: 4, line: 107-113)

Comment 3:

This study is about only 5 cases. No clear statistical analysis has not been done to show effect is significant and real. I think more number of patients study is needed to report this case study.

[Response]

Thank you for pointing this out. The number of cases in this paper is small, but this is due to the rarity of cases receiving this treatment as it is a treatment for a limited indication, as stated in the introduction.

Therefore, the following text has been added.

(1) We have accurately described this paper by stating the case in the title. “A Case Series on Pain Accompanying Photoimmunotherapy For Head and Neck Cancer” (page1)

(2) We have more clearly stated its weakness in the limitation as follows. “More data needs to be collected to clarify the results of this study in more detail, and prospective data collection should be considered in the future.” (Page: 8, line: 229-231)

For more details please see the revised version manuscript. 

Reviewer 3 Report

Major problems

1. In the maniuscript, Figure 1 provides detailed information on the structure, instrument parameters, and application scenarios of the frontal and cylindrical diffusers. However, the maniuscript text does not provide a comprehensive description of these two types of instruments. To improve the clarity and completeness of the paper, it is recommended that the authors provide a more detailed description of these two instruments, in conjunction with Figure 1. This would help readers better understand the instruments and enhance the readability and practicality of the paper.

2. In section 2.1 (Participants), It is recommended that the author move the information on the patients' clinical characteristics from section 3.1 to section 2.1 for a better description of the study subjects' features.

3. In Figure 2, the NRS curve associated with the use of the cylindrical diffuser for treatment shows that, theoretically, the NRS value should decrease gradually from POD0 to POD4. However, there is a significant increase in the NRS value at POD4, which requires a reasonable explanation from the authors.

4. Multi-spectral technology and hyperspectral imaging technology also have applications in cancer detection, such as references ‘Open-source mobile multispectral imaging system and its applications in biological sample sensing’ and ‘Smartphone imaging spectrometer for egg/meat freshness monitoring’. Please discuss the application of multispectral and hyperspectral technology in cancer detection

Minor Problem

5. The format of the captions in this manuscript appears to be problematic, and it is suggested that the authors adjust the caption format to comply with academic writing standards.

Author Response

Comment 1:

In the maniuscript, Figure 1 provides detailed information on the structure, instrument parameters, and application scenarios of the frontal and cylindrical diffusers. However, the maniuscript text does not provide a comprehensive description of these two types of instruments. To improve the clarity and completeness of the paper, it is recommended that the authors provide a more detailed description of these two instruments, in conjunction with Figure 1. This would help readers better understand the instruments and enhance the readability and practicality of the paper.

[Response]

Thank you for your advice. The following text has been added.

“As tumor illumination methods, cylindrical diffusers placed in needle catheters are used to treat interstitial tumors, while frontal diffusers are used to treat superficial tumors. The non-thermal red light is applied to the tumor using a frontal diffuser for superficial light illumination for tumors ≤1 cm from the skin or mucosal surface or a cylindrical diffuser for interstitial illumination for tumors >1 cm from the skin or mucosal surface. The illumination time for frontal and cylindrical diffusers is 5 min for each treated area. For interstitial illumination, cylindrical diffusers are placed uniformly into the tumor 1.8 ± 0.2 cm apart using 17-gauge closed-tipped needle catheters under radiographic or ultrasound imaging.” (Page: 2, line: 48-56)

Comment 2;

In section 2.1 (Participants), It is recommended that the author move the information on the patients' clinical characteristics from section 3.1 to section 2.1 for a better description of the study subjects' features.

[Response]

Thank you for your advice. The following text has been corrected as you suggested. “Additionally, three of the five eligible patients received PIT more than once. Of these patients, one patient received a first PIT at another hospital, and one patient received the first PIT as part of the Phase I clinical trial, so the first data for these two patients were excluded from the analysis. Hence, we evaluated total of nine PIT sessions, which received PIT at the National Cancer Hospital East as daily clinical practice, and the number of times PIT was performed in each of the five cases: once, twice, and three times in two, two, and one patients, respectively. Regarding the illumination technique, two patients received treatment via frontal diffuser only, one via cylindrical diffuser only, and the remaining two via both techniques.”

(Page: 3, line: 91-100)

Comment 3:

In Figure 2, the NRS curve associated with the use of the cylindrical diffuser for treatment shows that, theoretically, the NRS value should decrease gradually from POD0 to POD4. However, there is a significant increase in the NRS value at POD4, which requires a reasonable explanation from the authors.

[Response]

Thank you for your advice. The following text has been added in result.

“The NRS decreased slightly until POD3. However, Cylimdrical subsequently showed an increase in NRS at POD4. This increase in NRS is not significantly different from the NRS of POD1-3 and is similar to that before hospitalization.” (Page: 4, line: 144-147)

The following text also has been added in discussion.

“NRS decreased slightly to POD3 for both frontal and cylindrical diffusers.

However, the NRS increased again at POD4 in the cylindrical diffuser. This increase in the NRS for POD4 was similar to baseline and may have been a patient-reported bias, as opioid use was also decreasing. Also, while NRS tended to be lower for treatment with a frontal diffuser than with a cylindrical diffuser, opioid pain management was still necessary, suggesting that opioid pain management is essential for PIT.” (Page: 7, line: 199-205)

Comment 4:

Multi-spectral technology and hyperspectral imaging technology also have applications in cancer detection, such as references ‘Open-source mobile multispectral imaging system and its applications in biological sample sensing’ and ‘Smartphone imaging spectrometer for egg/meat freshness monitoring’. Please discuss the application of multispectral and hyperspectral technology in cancer detection.

[Response]

The "multi-spectral technology" you suggested may be useful for cancer detection, but our paper is not about cancer detection, but about the reality of pain and its management in PIT. I tried to add the references you suggested later in the discussion and introduction, but they did not follow the logic of the paper. Therefore, we did not add the suggested references this time.

Comment 5;

The format of the captions in this manuscript appears to be problematic, and it is suggested that the authors adjust the caption format to comply with academic writing standards.

[Response]

As you pointed out, the captions were not written properly, and each has been revised. (Figure 1-3 and table 1-4)

Round 2

Reviewer 2 Report

In Figure 2 and Figure 3 the statistics analysis with standard error mean need to be done and should be there in graph.

Author Response

In Figure 2 and Figure 3 the statistics analysis with standard error mean need to be done and should be there in graph.

(response)

Thank you for your advice.

We have added standard error mean values to the graph in Figure 2 and Figure 3.

For more details please see the revised version manuscript.